# Leptin—A Potential Bridge between Fat Metabolism and the Brain’s Vulnerability to Neuropsychiatric Disorders: A Systematic Review

**DOI:** 10.3390/jcm10235714

**Published:** 2021-12-06

**Authors:** Gilmara Gomes de Assis, Eugenia Murawska-Ciałowicz

**Affiliations:** 1Department of Molecular Biology, Gdansk University of Physical Education and Sports, 80-336 Gdansk, Poland; 2Laboratory of Behavioral Endocrinology—BELab—Brain Institute, Federal University of Rio Grande do Norte, Natal 59078-970, Brazil; 3Department of Physiology and Biochemistry, University School of Physical Education, 51-612 Wroclaw, Poland; eugenia.murawska-cialowicz@awf.wroc.pl

**Keywords:** body fat mass, energy expenditure, adipokines, eating disorders, mental health

## Abstract

Background: Obesity and being overweight have been described as potential causes of neurological disorders. Leptin, a peptide expressed in fat tissue, importantly participates in energy homeostasis and storage and has recently been identified for its signaling receptors in neuronal circuits of the brain. Aim: To elucidate whether the endogenous modulation of leptin can be a protection against neuropsychiatric disorders. Method: A systematic review was performed in accordance with the PRISMA-P method, and reports of studies containing data of leptin concentrations in healthy individuals with or without obesity were retrieved from the PubMed database, using the combinations of Mesh terms for “Leptin” and “Metabolism”. Results: Forty-seven randomized and non-randomized controlled trials, dating from 2000 to 2021, were included in the qualitative synthesis. Discussion and conclusions: Leptin secretion displays a stabilizing pattern that is more sensitive to a negative energy intake imbalance. Leptin levels influence body weight and fat mass as a pro-homeostasis factor. However, long-term exposure to elevated leptin levels may lead to mental/behavioral disorders related to the feeding and reward systems.

## 1. Introduction

The implications of weight control for the development of metabolic disorders, such as cardiovascular diseases and diabetes, have been widely described in the scientific literature [1]. Genome-wide association studies have identified more than 100 genes involved in overweight and obesity phenotypes, and the complex polygenic gene-behavior interaction explains the relevance of exogenous factors in such processes [2,3,4,5,6,7]. Nevertheless, the identification of genetic components involved in energy metabolism does not account for individual needs regarding diet and/or exercise in order to achieve weight control [8,9].

From a physiological perspective, a peptide hormone expressed in adipocytes, named leptin, has been suggested to function as a regulator of energy storage/expense by acting as a physiologic translator of changes in circulating glucose (GLU) levels into electrical/chemical signals in the brain [10,11,12,13]. In the brain, leptin signals exclusively through its long receptor isoform (LepRl) and participates in a number of neuroendocrine systems: for instance, sexual behavior through leptin receptor at kisspeptin neurons [14].

There are short leptin receptor isoforms present to varying degrees in almost all tissues; alteration in leptin concentrations might be associated with the GLU uptake in muscle and brown adipose tissues, and in hepatic GLU production. Furthermore, the modulatory role of leptin in the innate and adaptive immune responses implies that the regulation of leptin levels may ameliorate multiple inflammatory conditions [15,16].

Experimental research reports a selective downregulation of the LepRls, which are expressed only in a subpopulation of neurons in some hypothalamic and extra-hypothalamic regions of the central nervous system (CNS), due to high concentrations of circulating leptin [17,18,19,20]. Clinically, individuals with obesity and increased levels of leptin have been characterized as leptin-resistant; meanwhile, the loss of leptin-LepRl signaling in the brain has shown to be sufficient to promote obesity [21,22,23].

More recently, the influence of metabolic stressors on leptin secretion has become an important focus of research on feeding behaviors and mental disorders [24]. The fine-tuning of leptin-LepRls signaling, which can be affected by either too-low or too-high leptin concentrations, appears to be crucial to ensure healthy eating behaviors [25,26,27,28]. Regulation of LepRls might be the linking factor enhancing the influence of weight control on the development of psychiatric disorders [29,30,31].

Most studies address leptin as a potential pharmacological therapy in the control of weight gain and obesity [32,33]. However, the complexity of leptin actions and its differential interactions with other metabolic factors, both in the body and in the brain, requires a deeper understanding of the physiology of leptin in terms of weight gain/loss control [34,35,36,37].

Little is known about the physiology of leptin concentrations in the wider population since studies most likely only address individuals displaying eating and weight disorders and associated diagnoses. For this reason, whether endogenous leptin can be efficiently modulated still requires discussion. Therefore, we performed a systematic review and retrieved all the available clinical studies on leptin in the past 20 years, addressing populations with or without obesity in cohorts free of associated diagnoses, and synthesized the current knowledge on leptin physiology in the health and prevention of metabolism-related mental disorders.

## 2. Methods

In accordance with the preferred reporting items for systematic review and meta-analysis protocols (PRISMA-P) [38], to access the available literature presenting data on leptin levels (either expression or circulating concentration) in basal metabolic conditions, a double-blinded search was conducted in the PubMed and Google scholar databases and ResearchGate suggested research, concerning the following PICO strategy: P—healthy, active or sedentary individuals, with or without obesity; I—sleep and/or diet regimes; C—healthy controls; O—leptin and/or leptin receptor data. For this purpose, combinations of the following Mesh terms were used in an advanced search of the PubMed platform: “Leptin” AND “Metabolism” OR “Receptors, Leptin” AND “Metabolism”. Searches were filtered for control trials and randomized control trials, and duplicates were removed. Google Scholar and ResearchGate were also assessed. Papers referring to SARS-CoV2 or COVID-19 were excluded. The retrieved studies, dating from 2000 to January 2021, were screened against the abstracts for inclusion criteria in a double-blinded manner.

One thousand and fifteen papers were retrieved, based on combinations of the search terms, and were screened by titles after the removal of duplicates. Four hundred and sixty-five studies were screened by titles and abstracts. The inclusion criteria comprised both controlled and randomized controlled clinical trials that presented data on leptin levels in healthy subjects with or without obesity, aged over 18 years old, and undergoing dietary and/or sleep interventions. Reviews, cohorts, brief communications, studies involving menopausal and pregnancy periods, drugs, and supplementation, as well as those addressing subpopulations with pathological conditions, were excluded. Forty-seven studies were included in the qualitative synthesis. A summary of the most frequently studied conditions and a search flowchart of the exclusion criteria are presented in Figure 1. A summary of the included studies metadata is presented in Table 1 and Table 2.

## 3. Results

The study by Vgontzas et al. [39] showed that in men with obesity, those with sleep apnea have higher leptin levels than those without apnea. Guerci et al. [40] demonstrated that levels of leptin are not influenced by fat intake and that women display higher levels of plasma leptin, regardless of their body weight.

Comparing the effects of diet and exercise on leptin levels in men with obesity undergoing a weight loss regime, Thong et al. [61] observed that leptin levels only decreased in those who either lost weight or changed their body composition. Furthermore, changes in leptin levels correlate with changes in total, subcutaneous, and visceral adipose tissue. Next, by comparing eucaloric and hipper vs. hypo-energetic diets in subjects without obesity, Chin-Chance et al. [62] reported that leptin levels are associated with ad libitum feeding. They showed that hypo-energetic diets decrease leptin levels, while hyper-energetic diets increase leptin levels. It was also found that leptin levels do not immediately return to their previous concentrations when individuals return to a eucaloric diet. Similarly, a study by Assali et al. [41] described an 18% decrease in leptin levels, coupled with insulin resistance and triglycerides, in subjects with obesity during an isoenergetic washout period between two hypo-energetic diets.

In a study by Herrmann et al. [63], subjects not suffering from obesity on diets of different fat and glycemic profiles showed no differences in their basal leptin levels and no effects of OGTT on their leptin responses. In a study by Tsai et al. [42], women without obesity who were either on a diet or exercise regime had similar changes in their leptin levels: a 27% and 32% fall after diet and exercise, respectively, that recovered with energy repletion. By performing postprandial thermogenesis tests using different meal configurations in women with and without obesity, Tentolouris et al. [43] observed that fasting energy expenditure correlated with leptin levels.

Matsumoto et al. [44] monitored heart rate variability in women with and without obesity, to analyze leptin interactions with the autonomic nervous system. Reduced sympathetic responsiveness to leptin variations suggested that peripheral leptin resistance was present in women with obesity. Kassab et al. [45] checked daytime leptin levels in women without obesity during Ramadan and found that prolonged fasting with interrupted nocturnal eating was associated with the elevation of both leptin and insulin levels.

When measuring arterial versus venous concentrations of leptin, Eikelis et al. [46] showed that leptin levels are higher in men with obesity and that the brain contributes over 40% of the whole-body leptin release. Baratta et al. [47] revealed that plasma adiponectin, but not leptin, was correlated with insulin sensitivity and triglycerides in individuals undergoing a long-term weight-loss program. They identified an increase in plasma adiponectin and a decrease in plasma leptin after the program. Weigle et al. [48] analyzed metabolic and appetite parameters in individuals with obesity who were following isoenergetic diets with different protein/fat profiles and concluded that spontaneous energy intake and mesor leptin levels decreased with an ad libitum high-protein diet, despite an increase in ghrelin levels.

Analyzing coronary blood flow in individuals without obesity by ergospirometry and echocardiography, Kiviniemi et al. [49] observed that leptin levels correlated with waist-to-hip ratio, fat percentage, body mass index, LDL cholesterol, oxidized LDL, and apolipoprotein B. Moreover, subjects with subcutaneous adipose tissue had a stronger association with leptin concentrations than those with visceral adipose tissue. The molecular analyses conducted by Joosen et al. [50] revealed that changes in the expression levels of peroxisome proliferator-activated receptor γ1 and 2, and the aP2 transcriptional factor, were positively related to changes in plasma leptin in women without obesity who received a 50% increase in energy intake for two weeks. Poppitt et al. [64] found that in men without obesity, fatty meals had no effect on postprandial levels of ghrelin; however, they found no decrease in leptin levels. Neither ghrelin nor leptin was correlated with circulating insulin levels.

In a study by Chapelot et al. [51], men without obesity who subtracted a meal from their standard regimen showed an increase in body fat mass and leptin levels, while those who added a meal showed no effect. In Bray et al. [65], men with obesity who were fed fast-food, organic, and meat-based meals had an increase in ghrelin levels only after fast-food ingestion. No changes were reported for leptin levels over a six-hour postprandial period. Carlson et al. [66] reported that fasting levels of insulin, leptin, ghrelin, adiponectin, resistin, and BDNF were not affected by feeding frequency, comparing one and three meals per day for two months. In Abete et al. [67], individuals with obesity who underwent a two-month fish-based hypo-energetic diet had a decrease in leptin levels that was positively correlated with decreased insulin.

Bouhlel et al. [52] investigated the effect of a four-week Ramadan period (fasting from 5 a.m. to 6 p.m.) on leptin levels in men without obesity. Ramadan fasting provoked a reduction in body mass and body fat mass, with no changes being recorded in leptin or adiponectin levels.

Individuals with obesity on two different sleep regimes over two weeks, in a study by Nedeltcheva et al. [68], showed no difference between leptin and ghrelin levels. Leptin concentrations increased in a similar fashion at the end of each shorter or longer sleep regime. Bortolotti et al. [69] demonstrated that either hypercaloric high-fat or hypercaloric high-fat high-protein diets increased plasma leptin concentrations in men without obesity. In research conducted by Ratliff et al. [70], men with obesity on a low-carb diet who received additional dietary cholesterol showed reductions in fasting insulin and leptin levels that correlated with reductions in body fat mass.

In a study by J. A. Cooper et al. [71], men with obesity undergoing a three-day eucaloric overfeeding or underfeeding diet showed a decrease in the levels of leptin during the underfeeding diet. Leptin levels increased during overfeeding only when individuals were first underfed.

In another study by Bhutani et al. [53], individuals with obesity, enrolled in an alternate-day fasting controlled feeding regimen versus alternate-day fasting self-selected feeding, showed a reduction of 21% and 23% in leptin and resistin levels, respectively. Lower triacylglycerol levels were associated with increased adiponectin and reduced leptin levels. In research conducted by Bergouignan et al. [54], after two months of sedentary vs. exercise regimes, women without obesity showed no difference in leptin levels. Their fasting leptin levels were negatively correlated with their spontaneous feeding. Wu et al. [55] analyzed adiponectin, leptin, and bone turnover markers in a large sample of women without obesity and revealed that adiponectin and leptin were the main peptides secreted by adipose tissue.

Individuals not suffering from obesity who committed to an 8-week weight gain followed by an 8-week weight loss regimen for a study by Adachi et al. [72] showed that weight gain is associated with increases in insulin and leptin concentrations. Moreover, immediate weight loss decreased their levels of insulin, leptin, and adiponectin toward baseline values. According to Lecoultre et al. [73], individuals with obesity showed a 44% decrease in the mesor leptin curve after six months of caloric restriction, although the diurnal amplitude of the curve slightly increased over that period.

Individuals with obesity undergoing both a high-fat (60% kcal) and a low-fat (25% kcal) diet with 25% energy restriction for a study by Varady et al. [74] showed decreases in leptin levels (low-fat, 48% vs. high-fat, 28%) and body weight after 6 weeks. Decreases in body fat mass, as well as increases in plasma adiponectin levels, were noted only for the low-fat diet. Benedict et al. [56] showed that in men without obesity under regular sleep vs. sleep deprivation conditions, metabolic turnover demonstrated a pronounced diurnal rhythm, peaking during the early afternoon, in contrast to leptin profiles that peaked early in the night.

Subjects with obesity who were already in the course of weight loss (10–15%) enrolled in either an isocaloric low-fat high-glycemic, a moderate-fat low-glycemic, or a high-fat very-low-glycemic diet for the study by Ebbeling et al. [75]. This revealed that serum leptin was highest with the low-fat diet, intermediate with the low–glycemic diet, and lowest with the very low-carbohydrate diet.

In the study by Chang et al. [76], individuals with and without obesity undergoing 4 weeks either on an isocaloric low- or a high-glycemic diet showed no difference in their leptin levels afterward. In another study by Varady et al. [77], individuals with obesity who submitted to a three-month alternate daily fasting regimen showed a decrease in their body weight and leptin with an increase in adiponectin levels. Cooper et al. [57] found that subjects with obesity who were undergoing a one-year weight loss regimen with diet and exercise reported decreases in aldosterone levels associated with decreases in leptin and insulin, and with increases in adiponectin levels.

Calvin et al. [78] demonstrated that sleep restriction increased caloric intake in individuals with obesity. Sleep restriction was not associated with changes in energy expenditure, leptin or ghrelin levels. Dos Santos Moraes et al. [58] observed a negative correlation between leptin and adiponectin/leptin ratios, which increased and decreased in individuals with obesity, according to the degree of obesity.

Müller et al. [59] reported that men without obesity, submitting to overfeeding followed by a caloric restriction diet and then refeeding, showed a reduction in fat mass during caloric restriction that increased with energy expenditure, leptin, insulin, adiponectin, T3, and testosterone. In those individuals with obesity in a weight-loss program researched by Williams et al. [60], women had higher basal levels of leptin, adiponectin and ghrelin, and the ratios of leptin to adiponectin and leptin to ghrelin. In women, higher ghrelin concentrations predicted greater weight loss and their higher leptin to ghrelin ratio predicted their weight gain within 6 months.

Singh et al. [80] demonstrated that in subjects without obesity who enrolled in a weight gain program, weight gain but not weight maintenance initiates an increase in adiponectin levels. Changes in adiponectin levels positively correlated with changes in leptin levels. In parallel, an in vitro experiment showed that leptin increases adiponectin production in white preadipocytes, while the leptin antagonist has the opposite effect. Comparing normal and obese tissue suggests that the leptin signaling pathways increase adiponectin expression in normal-weight but not in obese adipose tissue.

Perrigue et al. [81] investigated the effect of eating frequency on metabolic markers in subjects without obesity. Meals/day frequencies were negatively correlated with serum IGF-1 levels, but no associations were found for leptin levels. Individuals with obesity undergoing a two-year weight-loss program for the study by Liu et al. [82] reported that within a six-month to a two-year program, a decrease in free T3 and total T3 was positively associated with changes in their body weight, leptin, GLU, insulin, and triglyceride levels. Kessler et al. [83] explored the effects of two daily profiles of isocaloric diets in men without obesity on their metabolic factors. They reported that the diurnal patterns of adipokines were modulated by dietary patterns. Mesor leptin was lower when a carbohydrate-rich meal was consumed up until 1:30 p.m., with fat-rich meals between 4:30 p.m. and 10:00 p.m.

In the study by Trepanowski et al. [84], individuals with obesity who committed to a five-month period of either alternate-day fasting or caloric restriction showed no difference in their ratios of visceral to subcutaneous fat, their total mass ratios similarly increased in both diets, and their leptin levels decreased by 18% and 31%, respectively. In a study conducted by Hołowko et al. [85], men not suffering from obesity who enrolled in a six-week 20% or 30% hypo-energetic diet showed decreases in leptin levels, triglycerides and total lipids, with an increase in adiponectin levels.

## 4. Discussion

The participation of metabolites that are largely expressed in peripheral tissue such as cytokines (i.e., peroxisome proliferator-activated receptor γ co-activator 1α—PGC-1α, brain-derived neurotrophic factor (BDNF), and irisin) as active players in the etiology of neuropsychiatric diseases has highlighted metabolism as a relevant factor in research on mental health. It is now recognized that our everyday exercise and nutrition choices have long-term consequences on our brain function [86,87,88,89].

The identification of the central actions of leptin when signaling through its specific LepRls in the neurons of the neuroendocrine and hippocampal circuits elucidates a complex and integrated central control of body energy (expenditure vs. storage) and sophisticated self-defense of the brain, thus shedding light on how it modulates satiety and compulsion [30].

Evidence suggests that leptin and adiponectin are the major peptides secreted by adipose tissue, and that circulating levels of leptin positively correlate with body fat mass, particularly with subcutaneous adipose tissue. Leptin levels also correlate with the waist-to-hip ratio and BMI [44,49,54,55]. Leptin levels are higher in individuals with obesity compared to those without obesity [46], as well as in women compared to men [40].

Studies also confirm that leptin levels do not alter under fasting and/or postprandial conditions [43,64,65], whereas the mean levels of leptin seem to respond rapidly to changes in caloric intake (diet), both quantitatively and qualitatively. Specifically, leptin levels are more responsive to a negative energy balance regime, meaning that leptin secretion is more likely to decrease under hypocaloric diet conditions than to increase under hypercaloric diet conditions. Moreover, the levels of leptin appear to rapidly return to the standard starting point once eucaloric conditions are restored [53,59,61,70,71,72,75].

It is suggested that changes in leptin are independent of circadian rhythms and stress factors; however, the endogenous modulation of leptin might be essential in preventing various chronic metabolic and brain diseases [56,73,75,78,81]. Therefore, it is necessary to understand the physiology of leptin and a healthy metabolism. Changes in circulating leptin might be selectively responsive to fat stores as a source of energy nutrition, as opposed to glycemic indexes (commonly associated with satiety) that in turn have no impact on leptin secretion [63,75,76]. Furthermore, high-fat hypercaloric diets increase leptin levels independently of protein content [69]. Conversely, high-fat hypocaloric diets induce smaller decreases in leptin levels than low-fat diets [74].

From a behavioral perspective, however, Thackray et al. [90] found that high-fat hyperenergetic diets increase leptin levels and the appetite preference for high-fat over low-fat foods, despite reductions in ghrelin levels. Similarly, leptin levels decrease with spontaneous feeding under high-protein diet conditions, independently of increases in ghrelin levels [48,60,62]. This suggests that despite the stable feature of leptin, basal leptin levels might be responsive to short-term and long-term alterations in nutrition.

Both hypercaloric and hypocaloric diets modulate leptin levels in accordance with their fat content. While high-fat diets increase leptin levels and result in an increased appetite for high-fat foods, high-protein diets affect the appetite and feeding behaviors in such a manner that they downregulate leptin levels, due to reduced spontaneous feeding. This implies that the feeding control of body fat mass and obesity is ultimately linked to the downregulation and stabilization of leptin production.

Although leptin levels are not correlated with sleeping/waking regimes [56,68,78], sleep disorders related to obesity, such as sleep apnea, appear to be associated with increases in leptin levels [39]. Furthermore, diurnal leptin concentrations are affected by the time of day, regarding diet [51]. For instance, mesor leptin is lower when carbohydrate-rich meals are consumed before 1:30 p.m. and fat-rich meals are consumed between 4:30 p.m. and 10 p.m. [83].

In addition, adiponectin, known for its role in attenuating inflammation and regulating GLU levels and fatty acid breakdown, is also elevated in individuals with obesity [53,72,74,77,91,92]; it seems to have a positive metabolic counteraction against obesity and metabolic disorders. In line with this finding, subjects with obesity in a study by Dos Santos Moraes et al. [58] showed higher or lower concentrations of leptin/adiponectin according to their leptin levels and the degree of obesity. Later, Singh et al. [80] demonstrated that leptin activates the cellular signaling pathways that increase adiponectin expression in the adipose tissue of individuals without obesity.

These findings show that leptin levels seem to reflect inflammatory states in obesity, and that a reduction in leptin levels improves GLU metabolism, an effect that is more prominent in those who are overweight and obese [41,45,57,59,67,70,72,82]. This interplay of inflammatory cytokines in the control of body weight/composition makes leptin a pro-homeostasis stabilizer factor and might explain why it has not been successfully manipulated pharmacologically [93,94].

The involvement of central circuits in the expression of LepRls sheds light on the integration of metabolism as a contributing factor for feeding and stress-related neuropsychiatric disorders [95,96,97,98]. The function of leptin in the CNS requires fine-tuning of LepRls signaling throughout the neuroendocrine axes, which is a coadjutant in the physiopathology of these disorders. For instance, leptin suppresses the secretion of cortisol during the stress activation of the adrenal axis, thus contributing to the inhibition of chronic overactivation of the hypothalamic-pituitary-adrenal axis, whereas lowered leptin levels are associated with an increased risk of developing dementia [28]. Reduction in the energetic reserves and leptin production appear to cause a compensatory change in the reward system and the secretion of orexigenic and anorexigenic neuropeptides [99,100]; meanwhile, negative feedback in the expression of LepR1 as a response to high leptin concentrations has been suggested to occur as a state of leptin resistance and to result in an inability to detect satiety [86].

## 5. Final Considerations

Leptin secretion is more prone to be downregulated by the rate of energy intake, and diet can modulate leptin levels, both quantitatively and qualitatively. The stabilization of circulating leptin at high levels has long-term implications for weight levels and the feeding reward system, the stress-activation response, and the potential development of neuropsychiatric disorders of behavior.

## Figures and Tables

**Figure 1 jcm-10-05714-f001:**
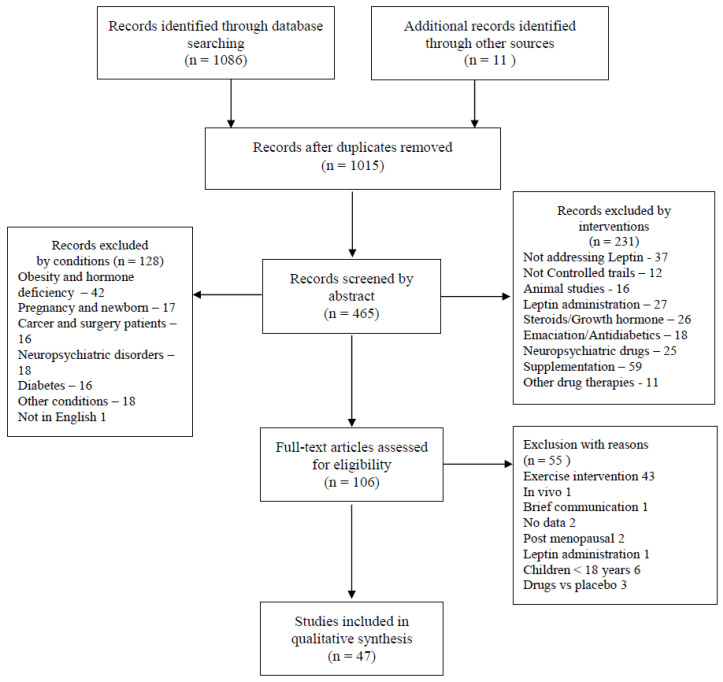
PRISMA search flow.

**Table 1 jcm-10-05714-t001:** Summary of non-randomized controlled studies’ metadata.

Non-Randomized Controlled Trials	Aim	Methods	Main Outcomes	Age Range	Fem (%)	BMI > 25	BMI < 25
Sleep apnea and daytime sleepiness and fatigue: relation to visceral obesity, insulin resistance, and hypercyt [39].	To test whether sleep apnea contributes to changes of tumor necrosis factor-α, interleukin-6, and leptin levels.	Obese, sleep apneic, and control subjects were investigated for leptin levels and other biochemical factors, and were monitored in the sleep laboratory for 4 consecutive nights.	Sleep-apneic men had higher plasma concentrations of leptin than non-apneic obese men or lean men, who had the lowest values.	38–49	0	11	12
No acute response of leptin to an oral fat load in obese patients and during circadian rhythm in healthy controls [40].	To elucidate the relationship between postprandial leptin and obesity, and the possible influence of the circadian rhythm on the dynamic leptin response	Leptin and insulin responses to an oral fat load test were measured according to the time of fat load ingestion: 0700 h (diurnal (D) test) or 2200 h (nocturnal (N) test) in 9 normal-weight healthy men.	Leptin concentrations were higher in non-obese women than men, and in obese subjects than in controls. No correlation was found between postprandial leptin and insulin. Leptin concentrations were not acutely influenced by a high fat intake load.	19–25	75	16	0
Insulin resistance in obesity: body weight or energy balance? [41].	To investigate the impact of diet on amelioration of the metabolic abnormalities associated with insulin resistance in obese subjects.	Twelve healthy obese subjects signed for a 6-week hypo-energetic diet, then a 4-week iso-energetic diet, then another 6-week hypo-energetic diet again. Insulin sensitivity, GLU tolerance and leptin levels were assessed.	One-third of the weight losses were achieved within the first week of diets. During the iso-energetic weight-maintaining period, insulin resistance decreased by 43%. Serum levels of leptin and triglyceride followed a similar pattern, but to a lesser extent.	38–57	50	12	0
Dieting is more effective in reducing weight but exercise is more effective in reducing fat during the early phase of a weight-reducing program in healthy humans [42].	To compare the effectiveness of food restriction and physical activity on body weight, composition and concentration of serum leptin in female subjects.	Thirteen non-obese subjects signed up for a two-phase crossover study including 9-day hypo-energetic treatments, either by food restriction or increased physical activity, with a 14-day washout in between and a 5-day follow-up energy repletion period. Parameters were established before and after the two phases.	Blood leptin levels were reduced by 27.02% and 32.27% after 9 days of hypo- energetic treatment—diet and exercise, respectively. The follow-up energy repletion increased the serum leptin concentration by ~45% in both cases.	22–55	100	0	13
Differential effects of high-fat and high-carbohydrate isoenergetic meals on cardiac autonomic nervous system activity in lean and obese women [43].	To compare the effects of two different isoenergetic meals on the sympathetic nervous system activity in lean and obese women.	Obese and nonobese subjects were examined after a CHO-rich and a fat-rich test meal. Blood pressure, heart rate, resting energy expenditure, plasma GLU, lipids, insulin, leptin, and norepinephrine were performed at baseline and every 1 h for 3 h after the meals.	Meal-induced thermogenesis was higher after CHO-rich, compared to fat-rich meals. CHO-rich meals caused greater cardiac SNS activation in non-obese than in obese women but SNS activation did not appear to influence the thermic effect of the food in either non-obese or obese women.	26–45	100	15	15
The potential association between endogenous leptin and sympathovagal activities in young obese Japanese women [44].	To investigate the association between leptin levels and sympathovagal activities in age- and height-matched obese and non-obese healthy young women.	Plasma leptin concentrations were measured by radioimmunoassay and autonomic nervous system activity was assessed during the rest period by a power spectral analysis of heart rate variability, and were identified as very low (VLO), low (LO), and high (HI).	The global SNS index to plasma leptin concentration was markedly reduced in the obese compared to the control group. Dynamics of BMI interaction suggested a reduced sympathetic responsiveness to endogenous leptin production in obese women, and that 30% of total body fat might be the critical point at which leptin resistance is induced.	18–22	100	15	15
Interactions between leptin, neuropeptide-Y and insulin with chronic diurnal fasting during Ramadan [45].	To describe changes and interactions between leptin, neuropeptide-Y, and insulin during the pattern of fasting in Ramadan.	Blood variables were measured at 1:00–2:00 p.m. during baseline and on days 14 and 28 of Ramadan fasting—abstinence from all food or drink, including water and chewing gum, from dawn to sunset.	Long-term fasting with interrupted nocturnal eating is associated with significant elevations in serum leptin and insulin and reduction in serum neuropeptide-Y. Changes in leptin appeared to be mediated by insulin. Neuropeptide-Y changes were independent of leptin or insulin.	20–24	100		46
Extra-adipocyte leptin release in human obesity and its relation to sympathoadrenal function [46].	To measure total and regional norepinephrine spillover, epinephrine secretion rate, and extra-adipocyte leptin release in lean and obese subjects.	Total and regional extra-adipocyte leptin release was measured in obese and nonobese men by arterial and central venous catheterization. Because the plasma clearance of leptin is primarily by renal removal, whole-body leptin release to plasma from renal plasma leptin extraction was estimated.	Whole-body leptin release was 1950 SE 643 ng/min in obese men and 382 SE 124 ng/min in non-obese men. A large contribution of brain leptin release to the plasma leptin pool was found (>40% whole body leptin release), with greater leptin release in obese than in non-obese men.	35–50	0	20	22
Adiponectin’s relationship with lipid metabolism is independent of body fat mass: Evidence from both cross-sectional and intervention studies [47].	To investigate the relation between plasma adiponectin and leptin levels, insulin sensitivity, and serum lipids in a cross-sectional study.	Plasma adiponectin and leptin levels, insulin sensitivity, and serum lipids were analyzed in 242 non-obese and obese subjects participating in a 6–12-month weight-loss program.	Plasma adiponectin, but not leptin levels, were correlated with insulin sensitivity, insulin sensitivity index, HDL cholesterol, and triglycerides. Plasma adiponectin increased and plasma leptin decreased after weight loss, and the changes were correlated with subjects’ BMI.	17–70	64	95	107
A high-protein diet induced sustained reductions in appetite, ad libitum caloric intake, and body weight despite compensatory changes in diurnal plasma leptin and ghrelin concentrations [48].	To test whether increasing the protein content while maintaining the carbohydrate content in the diet lowers body weight by decreasing appetite and spontaneous caloric intake.	Caloric intake, appetite, anthropometry, and leptin levels were measured in obese subjects undergoing a higher-fat, a lower-fat, and an ad libitum diet over 12 weeks. The AUC of plasma insulin vs. time, leptin, and ghrelin were measured.	Satiety was markedly increased with the isocaloric high-protein diet, despite an unchanged leptin AUC, and spontaneous energy intake decreased with the ad libitum high-protein diet, despite a decreased leptin AUC and increased ghrelin AUC. An increase in dietary protein intake from 15% to 30% of energy at a constant carbohydrate level produces a decrease in ad libitum caloric intake that appears to be mediated by increased leptin sensitivity in the CNS, resulting in weight loss.	27–62	63	19	0
Determinants of coronary flow velocity reserve in healthy young men [49].	To identify the risk markers for attenuated coronary flow velocity reserve that exist in healthy young men without evident atherosclerotic risk factors.	Echocardiography and ergospirometry sessions were conducted, and coronary blood flow was measured in non-obese men. Anthropometric measures and the regulation of fat metabolism were assessed by determining the adiponectin and leptin levels.	There was no relationship between coronary flow velocity reserve and serum lipids or body mass index. It was concluded that abdominal fat accumulation and low aerobic fitness are independently associated with coronary flow velocity reserve in men.	19–36	0	0	37
PPARγ activity in subcutaneous abdominal fat tissue and fat mass gain during short-term overfeeding [50].	To investigate whether initial subcutaneous PPARγ activity is related to fat mass generation during overfeeding.	Fourteen healthy non-obese subjects were overfed with a diet supplying 50% more energy than the baseline energy requirements for 14 days. Blood analyses of leptin, insulin and GLU, as well as mRNA expression of PPARg1, PPARg2, aP2 and UCP2, in fasting subcutaneous abdominal fat were performed.	Basal levels of Initial PPARg1 and 2, aP2 and UCP2 mRNAs were not related to fat gain. Changes in PPARg1, PPARg2 and aP2 mRNAs were positively related to changes in plasma leptin.	20–28	100	0	14
The consequences of omitting or adding a meal in males, regarding body composition, food intake, and metabolism [51].	To investigate the consequence on body composition and related biological and metabolic parameters of omitting or adding a meal.	Twenty-four nonobese subjects were recruited to either omit or add a fourth meal during a 28-day period; blood parameters were tested from lunch to the spontaneously requested dinner in each condition.	Omitting a meal was followed by increases in fat mass and late-evening leptin concentration. Adding a meal had no effect. In both groups, the change in energy content of the fourth eating regime was correlated with the change in adiposity.	19–25			23
Ramadan fasting’s effect on plasma leptin, adiponectin concentrations, and body composition in trained young men [52].	To evaluate the effect of Ramadan fasting on parameters of insulin resistance in trained athletes at rest and after aerobic exercise.	Nonobese subjects performed a progressive cycle-ergometer test one week before, after the first week, and during the fourth week of Ramadan, with biochemical analyses of GLU, cholesterol, HDL cholesterol, triglycerides, creatinine, and serum proteins, as well as insulin, leptin, and adiponectin.	Ramadan fasting was associated with a reduction in body mass and body fat without changes to leptin or adiponectin levels.	18–22	0		9
Improvements in coronary heart disease risk indicators by alternate-day fasting involving adipose tissue modulations [53].	To examine the effects of alternate-day fasting on adipokine profile, body composition, and coronary heart disease risk indicators in obese adults.	Obese subjects signed for a 10-week trial with the following consecutive diet phases: 2-week baseline control, 4-week alternate-day fasting with controlled feeding, and 4-week alternate-day fasting with self-selected feeding.	Leptin and resistin concentrations were reduced by 21 and 23%, respectively, after 8 weeks of treatment. Lower triacylglycerol concentrations were associated with augmented adiponectin and reduced leptin concentrations post-treatment.	40–52	75	12	0
Regulation of energy balance during long-term physical inactivity, induced by bed rest, with and without exercise training [54].	To determine the long-term effects of physical inactivity on energy balance regulation and to test the effect of exercise training on energy balance adjustment to physical inactivity.	Nonobese subjects were divided into two groups: 1—accomplished a strict 60-day bed rest, 2—a combined aerobic/resistance exercise training concomitantly with bed rest. Body composition, spontaneous energy intake, hunger, total energy expenditure, and fasting gut hormones were measured.	Leptin levels did not change in both groups and no group effect was observed. Leptin levels were strongly associated with fat mass. Fasting leptin levels, adjusted for fat mass, negatively correlated with the spontaneous energy intake after the program.	30–40	100	0	16
Relationships between serum adiponectin, leptin concentrations and bone mineral density, and bone biochemical markers in Chinese women [55].	To investigate whether serum adipocytokines concentrations are associated with bone mineral density and bone turnover markers.	Serum adiponectin, leptin concentrations, bone turnover biochemical markers, and bone mineral density were analyzed in 265 non-obese premenopausal women.	Adiponectin and leptin were the main circulating peptides secreted by the adipose tissue. There were no significant correlations between leptin concentrations and bone-specific alkaline phosphatase in premenopausal and postmenopausal women.	26–44	100	0	265
Monitoring the diurnal rhythm of circulating nicotinamide phosphoribosyltransferase (Nampt/visfatin/PBEF): impact of sleep loss and its relationship with glucose metabolism [56].	To examine the 24-h profile of the serum Nampt in humans under conditions of sleep and sleep deprivation, and to relate the Nampt pattern to morning postprandial GLU metabolism.	Non-obese subjects participated in two 24-h lab sessions starting at 1800 h, including either a regular 8 h per night sleep or continuous wakefulness. Serum Nampt and leptin were measured at 1.5- to 3-h intervals. In the morning, plasma GLU and serum insulin responses to standardized breakfast intake were determined.	Under regular sleep-wake conditions, Nampt levels displayed a pronounced diurnal rhythm, peaking during the early afternoon, which was inverse to leptin profiles, which peaked in the early night. When subjects stayed awake, the Nampt rhythm was preserved but the phase advanced by about 2 h.	20–24	0		14
Changes in serum aldosterone, associated with changes in obesity-related factors in normotensive overweight and obese young adults [57].	To verify whether reductions in serum aldosterone are associated with favorable changes in obesity-related factors in normotensive overweight/obese young adults.	Obese subjects in a clinical trial, examining the effects of a 1-year diet and physical activity intervention with or without sodium restriction on vascular health. Bodyweight, serum aldosterone, 24-h sodium and potassium excretion and obesity-related factors were measured at baseline, 6, 12 and 24 months.	Decreases in aldosterone were found to be associated with decreases in leptin, insulin, homeostasis assessment of insulin resistance, heart rate, tonic cardiac sympathovagal balance, and increases in adiponectin.	20–45	80	285	0
The role of leptinemia as a mediator of inflammation in obese adults [58].	To investigate the role of leptinemia, adjusted by tertiles, on the inflammatory state in obese adults, according to obesity degree.	Obese subjects assessed for inflammation markers such as leptin, adiponectin, and the plasminogen activator inhibitor were grouped according to the adjusted tertiles of the leptin levels.	A negative correlation between leptin concentrations and the adiponectin/leptin ratio, and a positive correlation with the leptin/adiponectin ratio. The ratios were decreased and increased, respectively, according to obesity degree.	30–58	72	43	0
Metabolic adaptation to caloric restriction and subsequent refeeding: the Minnesota starvation experiment revisited [59].	To address the variance and kinetics of adaptive thermogenesis, its associations with body composition in the context of endocrine determinants, and its effect on weight regain.	Non-obese subjects accomplished an over-feeding (1 week at + 50% of energy needs) followed by a caloric restriction (3 weeks at 50% of energy needs), and refeeding diet (2 weeks at +50% of energy needs). Adaptive thermogenesis and its determinants were measured, together with body composition, whole-body MRI, isotope dilution, and nitrogen and fluid balances.	Caloric restriction reduced fat mass and fat-free mass and led to reductions in energy expenditure, heart rate, blood pressure, creatinine clearance, the activity of the sympathetic nervous system, plasma leptin, insulin, adiponectin, T3, and testosterone. Sympathetic nervous system activity, plasma leptin, ghrelin, and T3 and their changes were not related to adaptive thermogenesis.	20–37	0		32
Energy homeostasis and appetite-regulating hormones as predictors of weight loss in men and women [60].	To characterize plasma leptin, ghrelin and adiponectin concentrations in overweight and obese males and females, and to determine whether baseline concentrations of these hormones predict weight loss.	Subjects were assessed for biochemical analyses pre- and post-3- and 6-month participation in a weight-loss program. Baseline concentrations of leptin, adiponectin and ghrelin were determined.	Females had higher baseline concentrations of leptin, adiponectin and ghrelin, as well as ratios of leptin:adiponectin and leptin:ghrelin. Additionally in females, a higher baseline total ghrelin predicted greater weight loss and a higher ratio of leptin:ghrelin predicted weight gain at six months. A higher leptin:ghrelin ratio is a predictor of weight loss failure in females.	18–60	55	119	0

CHO, carbohydrate; AUC, area under the curve; BMI, body mass index; PPAR, peroxisome proliferator-activated receptor.

**Table 2 jcm-10-05714-t002:** Summary of non-controlled studies’ metadata.

Randomized and Quasi-Randomized Controlled Trials	Aim	Methods	Main Outcomes	Age Range	Fem (%)	BMI > 25	BMI < 25
Plasma leptin in moderately obese men: independent effects of weight loss and aerobic exercise [61].	To describe the independent effects of weight loss and exercise on plasma leptin levels.	Fifty-two sedentary obese men were assigned for 12 weeks to one of the following conditions: (1) control, (2) diet-induced weight loss, (3) exercise-weight stable, and (4) exercise-induced weight loss. Diets consisted of 55–60% carbohydrate, 15–20% protein and 20–25% fat.	Plasma leptin and BMI did not change in the control or the weight-stable groups. Diet and exercise similarly decreased plasma leptin after weight loss. Exercise in the absence of weight loss did not alter leptin levels. Changes in leptin correlated with changes in total and subcutaneous, but not in visceral, adipose tissues.	42–48	0	52	0
Twenty-four-hour leptin levels respond to cumulative short-term energy imbalance and predict subsequent intake [62].	To determine whether leptin serves as an indicator of short-term energy balance by measuring the acute effects of small manipulations in energy intake on leptin levels in normal individuals.	Four consecutive dietary treatments of 3 days each were applied to 6 healthy non-obese subjects, and leptin levels were serially measured throughout the study.	During washout after the underfeeding period, leptin levels were 88% of the eucaloric baseline, compared with 135% following overfeeding. Leptin levels did not return to baseline with the washout diet and were restored only after cross-over to the complementary diet treatment. Ad libitum intake changes correlated with changes in leptin levels.	18–39	0	6	0
A high-glycemic-index carbohydrate diet alters the diurnal rhythm of leptin but not insulin concentrations [63].	To determine whether post-prandial and diurnal patterns of leptin levels were altered by the type of diet amount of fat.	Nine healthy non-obese adults signed for an 8-day regimen in one of the following isocaloric diets containing 15% protein: A—high glycemic index (GI) CHO, 30% fat; B—low GI CHO, 30%fat; C—high GI CHO, 20% fat; and D—low GI CHO, 20% fat. Serum GLU, insulin, and leptin levels were measured for 24 h on day 8, and on day 9 during an OGTT.	Diets with HGI altered the serum leptin diurnal pattern, causing a rise in leptin beginning at 13:00 h. The area under the curve for leptin between 12:30 and 24:00 h in these diets was 17% greater. During the OGTT, leptin concentrations were similar in all diets.	20–37	44		9
Effect of high-fat meals and fatty acid saturation on postprandial levels of the hormones ghrelin and leptin in healthy men [64].	To investigate whether high-fat meals, which differed in saturated fatty acid content, acutely modified these hormones.	Eighteen nonobese subjects experienced a high-fat test meal (71% of energy as fat) as breakfast with either a high or low saturated:unsaturated fatty acid ratio on two occasions. Fasting and postprandial measurements of ghrelin, leptin and insulin were performed over 6 h, and at 10 and 24 h following a fat-exclusion lunch, snack, and dinner.	There was no significant effect of fatty meals (either saturated or unsaturated) on ghrelin over 6 h. Leptin decreased in response to both high-fat meals. There was no correlation between ghrelin or leptin and circulating insulin. Increasing dietary saturated fatty acids had no deleterious effects on leptin or total ghrelin.	18–28	0		18
Hormonal responses to a fast-food meal compared with nutritionally comparable meals of different composition [65].	To compare differences in the acute metabolic response to fast food and “healthy” meals with a similar composition.	Obese subjects were given a standard breakfast followed by one of the three lunches at noon: a fast-food meal; an organic beef meal; and a turkey sandwich made with organic foods and an organic orange juice. Blood parameters were measured every 30 min over a period of 6 h.	Leptin levels, whether expressed as a percent of baseline or absolute values, showed no significant pattern of change over the 6-h postprandial period.	19–27	0	6	0
Impact of reduced meal frequency, without caloric restriction on glucose regulation in healthy, normal-weight middle-aged men and women [66].	To evaluate the influence of reduced meal frequency, without a reduction in energy intake, on GLU metabolism in normal-weight, healthy male and female subjects.	Nonobese subjects underwent two 8-week ad libitum diet periods for weight maintenance with either three meals/day (breakfast, lunch, and dinner) or one meal/day (during a 4-h period in the early evening; 4:00 to 8:00 p.m.), with an 11-week washout period. Energy metabolism was evaluated throughout the study by morning OGTT, GLU, insulin, glucagon, leptin, ghrelin, adiponectin, resistin, and BDNF levels.	Diet had no significant effects on morning plasma glucagon, leptin, adiponectin, resistin, and BDNF.	40–50	33	0	15
Specific insulin sensitivity and leptin responses to a nutritional treatment of obesity, via a combination of energy restriction and fatty fish intake [67].	To investigate whether the inclusion of three fatty fish servings per week within a hypocaloric diet may have specific health effects on insulin and leptin functions.	Obese subjects were assigned to a control, a fish or a fatty fish-based diet, with a −30% total energy expenditure caloric restriction over an 8-week period. Anthropometry, body composition, lipid profile, leptin and insulin values were measured pre- and post-intervention.	A decrease in leptin was only observed after the fish-enriched diets. There was a positive correlation between insulin and leptin decreases. Sixteen percent of the variability in leptin changes could be explained by the HOMA index change and the type of diet.	29–45	44	32	0
Sleep curtailment, accompanied by an increased intake of calories from snacks [68].	To test whether the curtailment of human sleep could promote excessive energy intake.	Obese subjects were confined in a sleep laboratory for 14-day stays with ad libitum access to palatable food and either 5.5-h or 8.5-h bedtimes. Calories consumed during each bedtime condition, total energy expenditure, and 24-h profiles of serum leptin and ghrelin were analyzed.	No significant differences were found in serum leptin and ghrelin between the two sleep conditions. Twenty-four-hour leptin concentrations increased in a similar fashion at the end of the 5.5- and 8.5-h bedtime interventions.	35–45	45	11	0
High protein intake reduced intrahepatocellular lipid deposition in humans [69].	To assess the effect of high protein intake on high-fat diet-induced IHCL accumulation and insulin sensitivity in healthy young men.	Non-obese subjects crossover after 4 days on a hypercaloric high-fat diet; a hypercaloric high-fat, high-protein diet; and a control, isocaloric diet. Fasting metabolism analyses were performed at the end of the fourth day of each period, and the expression of key lipogenic genes was assessed in subcutaneous adipose tissue biopsies.	The high-fat diet and hypercaloric high-fat high-protein diet significantly increased plasma leptin concentrations.	22–26	0		10
Carbohydrate restriction (with or without additional dietary cholesterol, provided by eggs) reduces insulin resistance and plasma leptin without modifying appetite hormones in adult men [70].	To evaluate the effects of additional dietary cholesterol and protein, provided by whole eggs while following carbohydrate-restricted diets, on insulin resistance and appetite hormones.	Obese subjects were allocated to the following 12-week regimes: egg (640 mg/d additional dietary cholesterol) or placebo (0 mg/d additional dietary cholesterol) while following carbohydrate-restricted diets.	There were significant reductions in fasting insulin and fasting leptin concentrations for both groups, which were correlated with the reductions in body weight and body fat.	40–70	0	31	0
Serum leptin levels in obese males during over- and underfeeding [71].	To test whether leptin responds to short-term changes in energy balance in obese and lean males.	Obese subjects were signed to four consecutive dietary treatment periods of 3 days each: eucaloric diet, followed by periods of overfeeding (130% of total energy expenditure) or underfeeding (70%), separated by eucaloric (100%) washout period. Total energy expenditure and leptin levels were measured during the third day of each treatment.	Leptin levels were acutely responsive to negative energy balance, but not to positive energy balance unless subjects were previously underfed. During overfeeding, leptin levels increased by 25 ± 11% when subjects were underfed first, but did not increase when subjects were overfed first.	20–35	0	8	0
Effect of weight gain on cardiac autonomic control during wakefulness and sleep [72].	To investigate the effect of fat gain on cardiac autonomic control during wakefulness and sleep in humans.	Non-obese subjects were engaged in either a gaining-weight diet (4 kg in 8 weeks) followed by an 8-week weight-loss period or a weight-maintainer control. Analysis of heart rate variability was performed at baseline, after weight gain, and after weight loss, to examine the relationship between changes in heart rate variability and changes in insulin, leptin, and adiponectin levels.	Weight gain was associated with increased insulin and leptin concentrations. Insulin, leptin, and adiponectin increased after fat gain and fell after fat loss.	26–32	38		36
The fall in leptin concentration is a major determinant of the metabolic adaptation induced by caloric restriction, independently of the changes in leptin circadian rhythms [73].	To explore the changes in 24-h leptin circadian rhythms in response to caloric restriction, and the relationship between these changes and metabolic adaptation.	Obese subjects were assigned to a control group or a caloric restriction group for 6 months. Leptin concentration was assessed every 30 min for 24 h, and leptin circadian variations were fitted by Cosinor analysis. Sedentary energy expenditure and urinary catecholamine excretion were measured for 24 h in a metabolic chamber.	A decrease in the 24-h mesor of circulating leptin of 44 ± 3% was observed. The diurnal amplitude of leptin slightly increased over the diet. A metabolic adaptation of −126 ± 25 kcal/d with a decrease in urinary norepinephrine and T3 concentrations was observed, independently of the changes in mesor leptin.	26–48	57	46	0
Improvements in vascular health by a low-fat diet, but not a high-fat diet, are mediated by changes in adipocyte biology [74].	To examine how modulations in flow-mediated dilation by high-fat and low-fat diets relate to changes in adipocyte parameters.	Obese subjects were randomized to a high-fat diet (60% kcal as fat) or a low-fat diet (25% kcal as fat) for 6 weeks. Both groups were restricted by 25% of their energy needs.	Bodyweight decreased in both groups. Fat mass and waist circumference were reduced in the low-fat group only (−4.4 ± 0.3 kg; −3.6 ± 0.8 cm, respectively). Increases in plasma adiponectin and decreases in resistin were shown by the low-fat diet only. Greater decreases in leptin were observed with low-fat (−48 ± 9%) vs. high-fat (−28 ± 12%) diets.	32–40	70	17	0
Effects of dietary composition on energy expenditure during weight-loss maintenance [75].	To examine the effects of three diets, differing widely in macronutrient composition and glycemic load, on energy expenditure following weight loss.	Obese subjects, after 10% to 15% weight loss in a diet, consumed an isocaloric low-fat diet (60% carbohydrate, 20% fat, 20% protein; high glycemic load), low–glycemic index diet (40% carbohydrate, 40% fat, and 20% protein; moderate glycemic load), and very low-carbohydrate diet (10% carbohydrate, 60% fat, and 30% protein; low glycemic load) in random order, for 4 weeks.	Leptin was highest with the low-fat diet, intermediate with the low-glycemic index diet, and lowest with the very-low-carbohydrate diet.	24–38	38	21	0
Whether a low-glycemic-load experimental diet was more satiating than a high-glycemic-load diet [76].	To investigate the effect of low and high glycemic load diets on satiety and BMI, sex, and serum leptin.	Non-obese and obese subjects were crossed in a feeding study, testing low- vs. high-glycemic-load diets for 28 days, with isocaloric macronutrient distributions differing only in glycemic load and fiber. Fasting leptin levels were measured after each of the diet periods.	Serum leptin concentrations did not differ after the two diet treatments.	19–42	50	42	40
Alternate day fasting for weight loss in normal weight and overweight subjects: a randomized controlled trial [77].	To examine the effect of alternate-day fasting on body weight and coronary heart disease risk in non-obese subjects.	Obese subjects were randomized to either an alternate day fasting—ADF; ad libitum “feed day”, alternated with 25% energy intake “fast day”—group or a control group for 12 weeks.	Bodyweight decreased in the alternate-day fasting group. Adiponectin levels increased while leptin decreased in the intervention group vs. controls. LDL cholesterol, HDL cholesterol, homocysteine and resistin concentrations remained unchanged.	42–50	73	30	0
Effects of experimental sleep restriction on caloric intake and activity energy expenditure [78].	To determine the effect of 8 days/8 nights of sleep restriction on caloric intake, activity energy expenditure, and circulating levels of leptin and ghrelin.	Obese subjects were randomized into usual sleep vs. a 2/3 sleep restriction for 8 days/8 nights in a hospital-based clinical research unit. Parameters of caloric intake, activity energy expenditure, and circulating levels of leptin and ghrelin were analyzed.	Caloric intake increased in the sleep-restricted group and decreased in the control group. Sleep restriction was not associated with changes in activity energy expenditure. No change was seen in the levels of leptin or ghrelin.	18–40	35	17	0
Small-sided games training reduces CRP, IL-6 and leptin in sedentary, middle-aged men [79].	To improve chronic systemic inflammation, which provides protection against the ensuing development of chronic disease.	Subjects were engaged in cycling or small-sided games 3 days/week for 8 weeks, or sedentary control conditions. Pre- and post- evaluations included a dual-energy X-ray absorptiometry scan, sub-maximal aerobic capacity (VO2) and fasting blood analyses of C-reactive protein, interleukin-6, IL-1β, tumor necrosis factor-α, and leptin; IL-10, IL-1 receptor agonist, and adiponectin.	Both treatments increased submaximal VO2 and decreased total body fat mass and C-reactive protein. Only small-sided games increased total body fat-free mass and the concentration of plasma IL-6 and leptin.	40–58	0	32	0
Differential effects of leptin on adiponectin expression with weight gain, versus obesity [80].	To investigate the role of weight gain, and the consequent changes in leptin, on altering adiponectin expression in humans.	Non-obese subjects were engaged to either gain 5% of body weight by 8 weeks of overfeeding or to maintain weight.	Modest weight gain (3.8 ± 1.2 kg), but not weight maintenance, resulted in an increased adiponectin level, positively correlated with changes in leptin. In vitro analyses showed that leptin activates cellular signaling pathways and increases adiponectin mRNA in normal-weight, but not in obese, adipose tissue. Obese subjects showed increased caveolin-1 expression, which attenuates leptin-dependent increases in adiponectin.	22–35	14	0	44
Randomized trial testing the effects of eating frequency on two hormonal biomarkers of metabolism and energy balance [81].	To verify the possible influence of eating frequency on fasting plasma insulin-like growth factor-I and leptin.	The effects of eating frequency on fasting plasma insulin-like growth factor-I (IGF-1) and leptin were observed in non-obese subjects engaged in two eucaloric interventions lasting 21 days each: low frequency (3 eating occasions/day) and high frequency (8 eating occasions/day).	There were lower serum IGF-1 levels when subjects underwent high- compared to low-eating-frequency conditions. There was no association between eating frequency and plasma leptin levels.	19–38	73		11
Thyroid hormones and changes in body weight and metabolic parameters in response to weight-loss diets: the “Pounds Lost” trial [82].	To examine the associations between thyroid hormones and changes in body weight and resting metabolic rate (RMR) in a diet-induced weight-loss setting.	Obese subjects from the 2-year “Pounds Lost” trial were assessed for body weight and resting metabolic parameters. Thyroid hormones (free triiodothyronine (T3), free thyroxine (T4), total T3, total T4, and thyroid-stimulating hormone (TSH)), anthropometric measurements, and biochemical parameters were assessed at baseline, 6 months, and 24 months.	Decreases in free T3 and total T3 levels were positively associated with changes in body weight and leptin, GLU, insulin, and triglycerides, both in 6 months and 2 years. No association was observed between baseline leptin levels and weight change.	30–70	83	569	0
Diurnal distribution of carbohydrates and fat affects substrate oxidation and adipokine secretion in humans [83].	To investigate the effects of dietary patterns on energy metabolism, and circulating lipids, adipokines, and inflammatory markers.	Non-obese subjects underwent two isocaloric 4-week diets: (1) carbohydrate-rich meals until 13:30 and fat-rich meals between 16:30 and 22:00; or (2) the inverse sequence of meals. During a 12-h clinical investigation day after each intervention period, two meal tolerance tests were performed, at 09:00 and 15:40, respectively. Substrate oxidation and circulating lipids, adipokines, and cytokines were assessed pre- and postprandially.	Diurnal patterns of triglycerides, LDL cholesterol, leptin, visfatin, and LPS-induced cytokine secretion in blood leukocytes were modulated by the diets. Average daily concentrations of leptin and visfatin were lower on Diet 2 than on Diet 1.	42–48	0		29
Effects of alternate-day fasting or daily calorie restriction on body composition, fat distribution, and circulating adipokines: secondary analysis of a randomized controlled trial [84].	To compare changes in the visceral adipose tissue:saturated adipose tissue ratio, fat-free mass:total mass ratio, and the adipokine profile between alternate-day fasting and daily calorie restriction.	Obese subjects were signed to (1) alternate-day fasting (alternating every 24 h between consuming 25% or 125% of energy needs); (2) calorie restriction (consuming 75% of needs every day); or (3) control (consuming 100% of needs every day) over 24 weeks.	Visceral adipose tissue:subcutaneous adipose tissue ratio did not change. The FFM:total mass ratio similarly increased in both groups compared to the control group. Circulating leptin levels decreased by 18% and 31%, respectively, in the diet groups relative to control. The levels of adiponectin, resistin, IL-6, and TNF-a did not change in either intervention group relative to control.	42–48	84	79	0
Six weeks of calorie restriction improved body composition and lipid profile in obese and overweight former athletes [85].	To compare the impact of 20% vs. 30% (of total daily energy expenditure) reduction of daily caloric intake over 6 weeks on body mass reduction and insulin metabolism in former athletes.	Obese subjects signed to 6-week of either a 20% or a 30% restriction diet had body composition variables, lipid profile (total lipids—TL; total cholesterol—TCh; HDL cholesterol—HDL; LDL cholesterol—LDL; triglycerides—TG), GLU, insulin, IGF-1, leptin and adiponectin.	A decrease in TG, TL and leptin, and an increase in adiponectin levels, were observed in both diet groups.	29–41	0	63	0

OGTT, oral glucose tolerance test; HGL, high-density lipoprotein; LDL, low-density lipoprotein.

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
