# Peer review of "Leptin—A Potential Bridge between Fat Metabolism and the Brain’s Vulnerability to Neuropsychiatric Disorders: A Systematic Review"

_jcm, 2021, doi:10.3390/jcm10235714_

Round 1

Reviewer 1 Report

The authors have been conducted an extensive search of the existing literature and have summarized some of the information related to leptin levels, energy consumption and some markers of mental health. I consider that the work has a lot of metabolic evidence but not enough evidence linking metabolic with markers of mental disorders. It is a good job; however I think the title does not match the evidence shown. Finally, I see the final considerations poor and without linking leptin levels, obesity markers and mental health markers.

Author Response

Dear reviewer,

Regarding the grammatic revision required and in order to properly attend the resourceful opinion of all the reviewers, we have made substantial adjustments in the text. We believe that the Title is now matching with the aim and discussion. And the final considerations are more precise.  

Reviewer 2 Report

 The review by de Assis and Murawska titled “Leptin – a bridge between overweight and mind disorders: A systematic review” scoped for relevant human studies to address the objective “are variations in leptin levels susceptible to endogenous modulation and may so protect from metabolism related disorder? Hence the authors performed a systematic review and retrieved from the past 20 years all the available clinical studies on leptin using obese and non-obese subjects without other conditions. Although such a review is important as a reference point for other future original work there are some discrepancies that will need to be addressed.

Title/ Abstract/ Introduction and Objective

  1. The title and objective should align better, either the title should be reworded to follow the objective or the objective should be written to match the title.

As the aims just states as metabolism related disorder but not specific to mind disorders.

  1. As stated in introduction and then again in discussion mind disorder can be grouped to sleep, lack of initiate to exercise; eating habits and so forth – in relation to metabolism, leptin and brain related triggers; however, once we get to the results and discussion it is not easy to follow the key summarise points. So once the results and discussion are corrected, the introduction will need to be checked so that the overall systematic review narrate a clear structure concept.

Methods

Methods are generally explained well but here are some questions to address in regards to this section:

  1. Please state why only one database was used?
  2. How were 11 papers from other sources identified i.e., using what method if not by database search?
  3. Keywords (Mesh or otherwise) were just in relation to leptin and its receptor and metabolism, yes? So only based on abstract the other disease that were not mind disorder related were excluded and lead to your 47 papers to investigate?
  4. So when you state healthy obese and non-obese patients can be misleading as obesity itself is a comorbidity that is not considered healthy…best to find a different term or better way to state this.

Results

  1. Table 1 is very long and can be divided into different tables

E.g., randomised studies and non-randomised studies (prospective observational, cross-sectional, retrospective etc)

  1. Then by findings and order the findings that are similar together as closely as possible
  2. Avoid repeat information within the table/s in the text and that will make it redundant

  1. The results section can have subheadings according the newer tables that will be created as suggested above, so more meaningful summary of findings can be made within this section.

Discussion

Once the results are reorganised as suggested above then the discussion points can be reorganised accordingly. Also, section as what is known now after your systematic review synthesis and what are still not clear and suggest how these gaps can be do address going forward.

So, the aim of this review is still unclear is it that you looking just at the metabolic level and mechanisms or are you looking at the level in accordance to intervention for subjects that are having mind disorders?

Even if this journal format for review does not allow subheadings your opening topic sentence per paragraph can divide the subsections clearly and make it easier to follow the evidence based narrative.

Thank you,

Author Response

Dear reviewer,

The title, aim presentation, as well as the whole introduction and discussion sections, have been revised in order to properly attend all the reviewers appointments and grammatic revision. Please have a look at new version.

  1. Please state why only one database was used?
  2. How were 11 papers from other sources identified i.e., using what method if not by database search?

Methods were corrected and the other sources of research were added, please find it in P.03 (line 98-99) 

3. Keywords (Mesh or otherwise) were just in relation to leptin and its receptor and metabolism, yes? So only based on abstract the other disease that was not mind disorder related were excluded and lead to your 47 papers to investigate?4. So when you state healthy obese and non-obese patients can be misleading as obesity itself is a comorbidity that is not considered healthy…best to find a different term or better way to state this.

Studies reporting populations with any diagnoses other than overweight were excluded from the results. This review, as opposed to previous publications, provides a synthesis of the physiology of leptin in non-disease conditions. We thus corrected the introduction, please check (lines 88-92) 

Results

  1. Table 1 is very long and can be divided into different tables E.g., randomised studies and non-randomised studies (prospective observational, cross-sectional, retrospective etc)
  2. Then by findings and order the findings that are similar together as closely as possible
  3. Avoid repeat information within the table/s in the text and that will make it redundant
  4. The results section can have subheadings according the newer tables that will be created as suggested above, so more meaningful summary of findings can be made within this section.

The table has been split into Non-RCT and RCT. The order of presentation of the studies's findings in the section Results was preserved as a timeline, in order to avoid reporting biases. Please check P.04 (line 126) and P.07 (line 128).

Discussion

Once the results are reorganised as suggested above then the discussion points can be reorganised accordingly. Also, section as what is known now after your systematic review synthesis and what are still not clear and suggest how these gaps can be do address going forward.

So, the aim of this review is still unclear is it that you looking just at the metabolic level and mechanisms or are you looking at the level in accordance to intervention for subjects that are having mind disorders?

Even if this journal format for review does not allow subheadings your opening topic sentence per paragraph can divide the subsections clearly and make it easier to follow the evidence based narrative.

Please have another look into the new version of the manuscript, regarding that it has been revised according to your valuable considerations and we hope it now fits your criteria.

Reviewer 3 Report

This is a good systematic review, following PRISMA recommendations, 

I have some comments:

-Title: I would change overweight by metabolism ( it was the word that you used for the search process and it implies that there is a relationship between overweight and leptin that you prove not to be really true). Besides it reflects better the major function of leptin in the organism

-Please include page numbers to be easier the communication with editors and reviewers

-I would recommend to show references in order of appearance and with numbers, not text

-Although you present and extensive review, you only use Pubmed as a searcher. PRISMA recommendations are to use 2-3 searchers motors for avoiding bias selections.

-Non-randomized control trials should be excluded from analysis

-Page 2 of introduction:

Line 50: "Influence multiple systems": for instance puberty through leptin receptor at kisspeptin neurons

line 78: "the consequent development..." I think this is sentence is simplistic. As you say later, it's a very complex system that regulates feeding with both central and peripheral signals.

line 84: "potential pharmacological therapies for obesity" : as you say later everyone failed to demonstrate it

line 91: "little is known". I miss Jean L Chan J Clin Invest 2003: "Role Leptin..." doi: 10.1172/JCI17490,  as a mother-paper of the knowledge that we have about leptin (most cited paper in this field)

-Figure 1. Flow diagram. You excluded 18 papers about neuropsychiatric disorders. If you try to relate leptin with mind disorders, shouldn't they be included?

-Results. Line 130. "Apnea have higher leptin levels". As you explain later leptin is related with body fat mass. I'm not sure if it can be confounding to explain each association without adjusting by the important one (fat mass)

-Line 164: if they are obese, they can't be healthy. Please rephrase

-Line 195. Leptin it's not related with insulin levels, it's not related with metabolic syndrome. Please, check through the manuscript this is clear.

-Discussion. Line 329. I would delete first sentence because it's irrelevant for your study. After that, I really enjoy your discussion that I found complete.

Line 423: "inflammatory state". Again leptin it's not related with insulin, with metabolic syndrome. Its association with inflammation seems coincidental with obesity

-Conclusions. Line458.459: From your study it can't be conclude that blood leptin is a good biomarker for metabolic disorders. I would delete this sentence.

Author Response

-Title: I would change overweight by metabolism ( it was the word that you used for the search process and it implies that there is a relationship between overweight and leptin that you prove not to be really true). Besides it reflects better the major function of leptin in the organism

According to the considerations of reviewers, the text has been fully revised in matter and grammar. References will be further edited.

The title was changed and we believe it to be more aligned with the aims and discussion. Please verify it.

-Although you present and extensive review, you only use Pubmed as a searcher. PRISMA recommendations are to use 2-3 searchers motors for avoiding bias selections.

This information was corrected, please check (lines 98-99). 

-Page 2 of introduction:

Please consider having another look at the whole Introduction due to the major revision provided in this version.

Line 50: "Influence multiple systems": for instance puberty through leptin receptor at kisspeptin neurons

Please verify correction in line 48

line 78: "the consequent development..." I think this is sentence is simplistic. As you say later, it's a very complex system that regulates feeding with both central and peripheral signals.

Please verify correction of full paragraph and line 78

line 84: "potential pharmacological therapies for obesity" : as you say later everyone failed to demonstrate it

The whole paragraph has been revised

line 91: "little is known". I miss Jean L Chan J Clin Invest 2003: "Role Leptin..." doi: 10.1172/JCI17490,  as a mother-paper of the knowledge that we have about leptin (most cited paper in this field)

This was a valuable recommendation, please check line 85

-Figure 1. Flow diagram. You excluded 18 papers about neuropsychiatric disorders. If you try to relate leptin with mind disorders, shouldn't they be included?

The main goal was to describe the physiology of leptin in a healthy organism and by so, elucidate the potential endogenous regulation of leptin in the protection of CNS disorders. So that, we have addressed the reports on individuals free of diagnoses other than overweight. We believe this information is now clarified in (lines88-92)   

-Results. Line 130. "Apnea have higher leptin levels". As you explain later leptin is related with body fat mass. I'm not sure if it can be confounding to explain each association without adjusting by the important one (fat mass)

In the Section Results, we have described the main outcomes reported in the included studies, free of reporting biases and following a timeline.  

-Line 164: if they are obese, they can't be healthy. Please rephrase

in accordance with a reviewer's recommendation, we are now using the terms "individuals with or without obesity" 

-Line 195. Leptin it's not related with insulin levels, it's not related with metabolic syndrome. Please, check through the manuscript this is clear.

We kindly ask you to have another look at the Manuscript Introduction and Discussion. We believe we have addressed your point. 

-Discussion. Line 329. I would delete first sentence because it's irrelevant for your study. After that, I really enjoy your discussion that I found complete.

Agreed. Correction in the line 316

Line 423: "inflammatory state". Again leptin it's not related with insulin, with metabolic syndrome. Its association with inflammation seems coincidental with obesity

The text has been corrected, please check line 388

-Conclusions. Line458.459: From your study it can't be conclude that blood leptin is a good biomarker for metabolic disorders. I would delete this sentence.

We have made the corrections, please verify line 423

Round 2

Reviewer 1 Report

The authors have performed extensive search of the existing literature and summarized some of the information related to leptin levels and its role as a molecule that can potentiate neuropsychiatric disorders of behavior. The article after a major review by the authors, the title and content based on evidence are read congruently, as well as the final conclusions. This review is accepted in the present form after minor revision. 

Minor Issues:

Line 448 Ethics approval: not applicable

Line 449 Consent for publication: not applicable

Line 450 Availability of data and material: not applicable

Line 451 Competing interests: the author declares no competing interests

Reviewer 2 Report

Thank you for addressing the comments provided. 

I have no further comments.